# Potential Application of an Aqueous Extract of *Tinospora Cordifolia* (Thunb.) Miers (Giloy) in Oral Submucous Fibrosis—An In Vitro Study

**DOI:** 10.3390/ma14123374

**Published:** 2021-06-18

**Authors:** Shankargouda Patil

**Affiliations:** Department of Maxillofacial Surgery and Diagnostic Sciences, Division of Oral Pathology, College of Dentistry, Jazan University, Jazan 45142, Saudi Arabia; dr.ravipatil@gmail.com

**Keywords:** giloy, oral submucous fibrosis, antifibrotic activity, *Tinospora cordifolia* (Thunb.) Miers

## Abstract

The in vitro antifibrotic activity of *Tinospora cordifolia* (Thunb.) Miers (giloy) was assessed to explore its potential for the management of oral submucous fibrosis. Epithelial cells dissociated from the tissue obtained from histopathologically normal oral mucosa during surgical extraction of third molars were cultured and fibrosis was induced by TGF-β1 in the oral keratinocytes. Cell viability was assessed by MTT and comparative gene expression analysis was carried out in the fibrosis-induced oral keratinocytes treated with various concentrations of *Tinospora cordifolia* extract (TcE) for matricellular protein-related gene expression. Concentrations of 0.5 µg/mL and 1 µg/mL TcE demonstrated a significant reduction in the expression of CTGF, SERPINE1, COL1A1, FN1, MMP1, MMP2, MMP3, and TIMP2 and an increase in the expression of PLAU, COL3A1, TIMP1, and TIMP3. Although TcE was found to reduce the expression of several fibrotic genes and increase the expression of antifibrotic genes, a varied effect was found, causing increased expression of COL3A1 and decreased expression of TIMP2 on TGF-β1-induced human buccal epithelial cells. However, further studies are warranted to assess the exact mechanism of antifibrotic activity and its clinical applications.

## 1. Introduction

Oral submucous fibrosis (OSMF) is one of the most common chronic conditions and potentially malignant disorders of the oral mucosa [1,2,3]. The disease is as old as 600 BC and has been described as “Vidari”, which means the progressive narrowing of the oral cavity, by Sushruta—the Father of Surgery [4]. In 1952, the term “atrophica idiopathic mucosae oris” was used to describe the condition [5]. In India, the first case of OSMF was reported in 1953 by Joshi [6]. Subsequently, there was extensive research on the condition, leading to much published literature in 1964, 1966, and 1968 by Pindborg, and a definition of the condition was formulated [7,8,9,10]. The incidence of the disease is higher in Asian countries, including India (the incidence rate in the rural population is 0.4%), Taiwan, Pakistan, Sri Lanka, China, and Bangladesh [3]. The disease is attributed to the use of “smokeless tobacco” by chewing, spitting, and snuff dipping. Paan and gutkha are chewed by individuals for mood elevation, alleviation of stress, its astringent properties, removal of germs, cleansing the oral cavity, and as a mouth freshener [11,12]. Children consume guthka due to its sweet taste, which, in turn, aids in microbial adhesion and decay.

The habit of consuming areca nut and smokeless tobacco correlates with the higher incidence of the disease reported in South Asian countries and the Southern states of India, such as Karnataka, Kerala, and Tamil Nadu [13]. The consumption of the raw form of areca nut is the result of increased commercialization of its products, which has, in turn, increased its exposure among young individuals since the early 1980s [14,15]. The International Agency for Research on Cancer has classified the areca nut as a group I carcinogen [16].

The nitrosamines and alkaloids of the areca nut, along with copper, cause OSMF by genotoxic activity [17,18,19]. When tobacco is chewed along with the areca nut, cholinergic receptors are activated, causing neuronal activation [20]. The increased salivary copper and iron due to areca nut also have deleterious effects, such as changes in the pH and flow rate of saliva [21]. Several studies have reported that the exposure of areca nut alkaloids to oral mucosa causes an imbalance in collagen metabolism, leading to the increased synthesis and decreased degradation of collagen [3]. Fibrogenic mediators, lymphokines, cytokines, growth factors, and hormones mediate collagen synthesis. Among the various cytokines, transforming growth factor-β (TGF-β) is predominantly associated with the pathogenesis of OSMF. It is a pleiotropic cytokine occurring in three isoforms, of which TGFβ1 is predominantly associated with OSMF [22]. It activates Smad2/3, leading to the transactivation of collagen type I and III [23]. The constant activation of TGF-β1 causes abnormal deposition and reduced degradation of types I and III collagen, leading to fibrosis [24]. Management for symptomatic improvement includes the use of collagenase, steroids, chymotrypsin, hyaluronidase, pentoxifylline, human placental extracts, vitamin and antioxidant supplements, surgical excision of the fibrotic bands, and laser ablation [25]. Yet, the disease is associated with a very high recurrence rate, and the aforementioned treatments aid only in providing temporary symptomatic relief. Repetitive intralesional injections cause discomfort to the patients and surgical therapy is associated with further aggression of fibrosis, worsening trismus [1]. Other alternative therapies include the use of hyperbaric oxygen, drugs possessing anti-inflammatory activity, and phytochemicals [26,27,28,29,30,31,32,33,34,35,36,37,38,39,40].

*Tinospora cordifolia* (Thunb.) Miers, commonly known as ‘Guduchi’ or ‘Giloy’, is found to grow in the hilly areas of India. The plant bears yellow-to-green-colored flowers and is a part of the Menispermaceae family [41,42,43]. The medicinal properties of this plant have been documented in the Indian Pharmacopoeia. It is an important constituent of the medicinal formulations for the treatment of fever, sexually transmitted diseases such as syphilis and gonorrhea, urinary tract infections, gout, anorexia, asthma, skin diseases, and viral hepatitis [44,45,46,47].

The phytochemical compounds that confer these medicinal properties include steroids, glycosides, phenolics, sesquiterpenoids, alkaloids, polysaccharides, diterpenoid lactones, and aliphatic compounds [47]. The antidiabetic, anti-inflammatory, and anticancer activity of giloy has been reported [48], but not its antifibrotic effect. With the available information, the present study was conducted to assess the in vitro antifibrotic activity of *Tinospora cordifolia* (Thunb.) Miers.

## 2. Materials and Methods

### 2.1. Collection of Tissue Specimen and Establishment of Primary Cell Culture of Human Oral Keratinocytes

The study protocol was approved by Scientific Research (IRB), College of Dentistry, Jazan University (CODJU-10213). Histopathologically normal oral mucosal tissues were obtained during the surgical extraction of third molars after obtaining informed consent from study participants (*n* = 5). The study used 0.25% trypsin-EDTA solution (Invitrogen, Carlsbad, CA, USA) to enzymatically digest tissue samples and prepare a single cell suspension. The epithelial cells that dissociated from the processed tissue were cultured with keratinocyte serum-free media (1×) (KSFM; Gibco, Gaithersburg, MD, USA) with 10% fetal bovine serum and incubated at 37 °C under 5% CO_2_ atmosphere. Cells from passage 1 were used for all the experiments.

### 2.2. Preparation of Aqueous Extract of T. cordifolia (TcE)

Finely ground powder of *T. cordifolia* stem was procured commercially from Dabur (India Ltd., Ghaziabad, India). Deionized water was added to the weighed powder and incubated for a day. The solution was constantly agitated until colorless water was obtained following overnight maceration, indicating completion of the extraction procedure. The insoluble particles were filtered through a syringe filter (0.22 micron-sized pores; Corning, NY, USA).

Leftover filtrate was lyophilized and a stock solution of 100 mg/mL was prepared by dissolving it in water and stored at 2–4 degrees Celsius until use.

### 2.3. Cell Viability Assay

Cell viability was measured using an MTT assay (3-(4,5-dimethylthiazol-2-yl)-2,5-diphenyltetrazolium bromide. In total, 5 × 10^3^ oral keratinocyte cells at passage 1 were seeded in each well of the 96-well plates. Cells were incubated for 24 h under ideal culture conditions to ensure cell attachment. Following this, various concentrations of TcE (0.5–25 µg/mL) were mixed with complete media. Further, the adhered cells were treated with appropriate concentrations of 1, 2.5, 5, and 10 µg/mL mixed with the complete KSFM with 10% FBS. Following a 48 h incubation in an incubator at 37 °C under 5% CO_2_, 0.5 mg/mL solution of MTT (Sigma-Aldrich Corp., St. Louis, MO, USA) was added to each well. This was incubated for 4 h at room temperature. Following the removal of the medium, 100 µL dimethyl sulfoxide (DMSO) (Sigma-Aldrich, St. Louis, MO, USA) was added to ensure the formation of formazan crystals, and the absorbance was measured at 570 nm with a Multiskan FC spectrophotometer (Thermo Scientific, San Jose, CA, USA).

### 2.4. Preparation of In Vitro Fibrosis Model

A total of 1 × 10^5^ oral keratinocytes were seeded in each well of 12-well plates and incubated for one day for cell adhesion. Fibrosis was induced by treating the cells with 10 ng/mL human TGF-β1 recombinant (R&D System, Minneapolis, MN, USA) for 2 h.

### 2.5. Assessment of Anti-Fibrotic Activity of TcE by RT qPCR

Cells were treated with selected concentrations of TcE (0.5 µg/mL and 1 µg/mL) mixed with the complete medium (KSFM + 10% FBS). Upon incubating for forty-eight-hours, cells were trypsinized and RNA was extracted with GeneJet purification columns (Invitrogen, Thermo Scientific, Vilnius, Lithuania). cDNA conversion of 1 μg of total RNA was performed with cDNA synthesis (High Capacity, Applied Biosystems, Carlsbad, CA, USA). Gene expressions of CTGF, PLAU, SERPINE1, COL1A1, COL3A1, FN1, MMP1, MMP2, MMP3, TIMP1, TIMP2, and TIMP3 were quantified with SYBRGreen PCR master mix (Applied Biosystems, Austin, TX, USA) on a QuantStudio 5 Real-Time PCR system (Applied Biosystems, Foster City, CA, USA). The housekeeping gene GAPDH was used to normalize the expression of the genes assessed in the experiment by ΔΔCt technique.

Data obtained from RT-PCR were quantified by calculating 2^−ΔΔCt^ values. A list of the primers (IDT, Coralville, IA, USA) is provided in Table 1.

### 2.6. Statistical Analysis

Two independent experiments were performed for every assay, and the results are presented as the mean ± standard deviation of the obtained values. Each test group was compared with the control (untreated cells) group using paired t-test (two-tailed) analysis using GraphPad Prism 8 (GraphPad Software, La Jolla, CA, USA). *p* < 0.05 was considered as significant and *p* < 0.01 was considered as highly significant (ns—not significant, * *p* < 0.05, and ** *p* < 0.01).

## 3. Results

### 3.1. Cell Viability Assay

The concentrations of 0.5–1 µg/mL of TcE treated keratinocytes revealed an increased number of viable cells compared with the control. At concentrations of 2.5, 5, and 10 µg/mL, the number of viable cells decreased in a dose-dependent manner. Since concentrations 0.5 and 1 µg/mL reduced epithelial toxicity, these concentrations were selected for the determination of antifibrotic activity (Figure 1).

### 3.2. Antifibrotic Activity of TcE by RT qPCR

Cells treated with TGF-β1 showed a significantly increased expression of CTGF (CTGF: CCN2 or connective tissue growth factor), SERPINE (PAI-1), COL1A1 (Collagen type I alpha-1) and FN1 (Fibronectin-1), MMP1 (Matrix metallopeptidase 1), MMP3 (Matrix metallopeptidase 3) and TIMP1 (Metallopeptidase inhibitor 1). A decreased expression of COL3A1 (Collagen type III alpha-1), TIMP2 (Metallopeptidase inhibitor 2), and TIMP3 (Metallopeptidase inhibitor 3) was observed in the cells induced with TGF-β1. An increased expression of MMP2 (Matrix metallopeptidase 2) and a decreased expression of PLAU (Plasminogen activator, urokinase (uPA) were observed, but the results were not statistically significant (data presented in Figure 2A–F, Figure 3A–F).

The TGF-β1-induced cells treated with 0.5 µg/mL or 1 µg/mL of TcE extract for 48 h demonstrated a significant decrease in the expression of CTGF, SERPINE1, FN1, TIMP2 and a decreased expression of MMP1, MMP2, MMP3, TIMP1, and TIMP3 in a dose-dependent manner in comparison with cells induced with TGF-β1 alone. A statistically significant increase in PLAU expression was observed on the TGF-β1-induced cells treated with 1 µg/mL of TcE compared with cells induced with TGF-β1 alone. Although there was a mild increase in PLAU expression at 0.5 µg/mL, the results were not significant. At 1 µg/mL, TcE induced a significant reduction in the expression of COL1A1 and increased expression of COL3A1 compared with TGF-β1-induced cells. Although similar results were observed at the 0.5 µg/mL concentration of TcE, the results were not significant (Figure 2A–F, Figure 3A–F).

## 4. Discussion

Oral submucous fibrosis is a collagen metabolic disorder caused predominantly by chewing areca nut with or without tobacco. The global estimate has revealed that the disease is more prevalent in Southeast Asian and Indian populations, with an approximate overall prevalence ranging between 0.2% and 0.5% [1]. This disease has been considered as a potentially malignant disorder owing to its transformation potential [2].

Considering the pathogenesis, areca nut chewing is associated with chronic inflammation, leading to the stimulation of fibroblasts, which results in the upregulation of various cytokines, such as TGF-β1, platelet-derived growth factor, connective tissue growth factor, and basic fibroblast growth factor. This affects various genes and proteins related to collagen metabolism, and ultimately causes decreased collagen degradation and increased collagen expression, causing oral submucous fibrosis. Areca nut and its compounds not only affect fibroblasts, but also epithelial cells. Early effects of the areca nut also modulate the exposure of these cells to neoplastic transformation.

Considering the treatment and prognosis of the disease, there is neither a specific cure, nor a change in the malignant transformation potential rate, despite several recent advances in therapeutic strategies. Thus, a therapeutic strategy that will aid in the management of the disease and prevent its malignant transformation is the need of the hour. In this regard, medicinal plant extracts are known to contain a mixture of phytochemicals that possess a combination of anti-inflammatory, immunomodulatory, and antioxidant potential that could exert antifibrotic activity.

With the available information, the present study was conducted to assess the effect of *Tinospora cordifolia* (Thunb.) Miers on TGF-β1-induced oral keratinocytes.

An MTT cytotoxicity assay carried out as a prerequisite investigation revealed that 2.5, 5, and 10 µg/mL concentrations of TcE extracts caused a significant impact on the viability of oral keratinocytes. The concentrations of 0.5 and 1 µg/mL reduced epithelial apoptosis and did not significantly affect cell viability, and were chosen as the precise concentrations for further investigation.

In the present study, the oral keratinocytes were induced with TGF-β1 prior to treatment with the extracts. Treatment of the cells with TGF-β1 caused several molecular and biochemical changes. Furthermore, there was a significant increase in the expression of connective tissue growth factor that plays a vital role in the pathogenesis of OSMF. It is a well-known fact that in the pathogenesis of areca nut-induced OSMF, the coarse particles of areca nut injure the oral mucosa and cause increased production of CTGF by oral cells at the sites of injury. CTGF has further been shown to induce collagen synthesis by both TGF-β1-dependent and -independent pathways. CTGF and TGF-β1, when acting in synchrony, are known to cause an intense fibrotic response [49].

Concerning the matrix metalloproteinases which are key enzymes in matrix degradation, the present study observed a significantly increased expression of MMP1 and 3 and also documented an increase in MMP2, although this was not statistically significant. This finding also deserves justification as MMPs are key molecules that control the degradation of collagen and various other matricellular proteins. Further, they also control the malignant transformation of OSMF lesions, as documented by some studies. Our present study results are concurrent with a study that has documented an increase in MMP1, 2, and 9 expressions in OSMF lesions [50]. A significant finding of another study that needs attention at this juncture is the fact that areca nut alkaloids induce senescence in oral fibroblasts and promote increased secretion of TGF-β and MMP2 that may develop a tissue environment crucial for the progression of OSMF to malignancy [51]. The increase in MMP levels documented in the present study could hint at the destructive influence of TGF-β on the malignant transformation of OSMF lesions mediated through MMP overexpression. The TIMP expression in our study following TGF-β treatment also deserves a mention.

The present study noted an increase in TIMP1 expression and a reduction in TIMP2 and 3 expressions, tipping the balance in favor of an overall reduction in TIMP expression. This finding is in disagreement with a study that has observed elevated TIMP1 and 2 expression in OSMF lesions [50].

The next significant observation in the present study is an elevation of SERPINE levels and a lowered expression of PLAU in TGF-β-induced keratinocytes, which is concurrent with similar studies that have established increased levels of protease activating inhibitors in OSMF lesions [52]. Significantly elevated COL1A1 and FN1 is another important finding in the TGF-β-induced keratinocytes, which are justifiable as OSMF is a disorder characterized by abnormal collagen deposition and reduced degradation. However, an intriguing finding of the present study was the reduction in COL3A1 expression following TGF-β.

The above findings encapsulate the baseline observations of the present study in the absence of giloy treatment. Upon administration of giloy extracts to the TGF-β-induced keratinocytes, many of the baseline findings were reversed and changed. The TGF-β1-induced cells on treatment with 0.5 and 1 µg/mL of TcE extract for 48 h demonstrated a significant decrease in CTGF, SERPINE1, FN1, and TIMP2 in a dose-dependent manner in comparison with cells induced with TGF-β1 alone. At 1 µg/mL, TcE induced a significant reduction in expression of COL1A1 in comparison with cells induced with TGF-β1-induced cells. The results obtained with giloy treatment implicate the antifibrotic role played by this herb in the pathogenesis of OSMF. Moreover, studies have reported that the extracts of this herb have been found to have antioxidant activity [53] and have been found to exert immunostimulatory effects [54]. There is also evidence that giloy extracts could inhibit NF kappa B expression [55], all of which depict the antifibrotic activity of the herb. However, the limitations of the present study are the fact that it is an in vitro study. Animal studies and human clinical trials are required after a thorough literature review and assessment of the safety profile of giloy.

## 5. Conclusions

So far, no study has highlighted the role of giloy in the management or cure of fibrosis. In this regard, the present study is novel. However, it has to be justified how giloy exerts its antifibrotic properties. It appears from a thorough understanding of the published literature that giloy exerts anti-inflammatory and immunomodulatory functions. By all these functions, giloy could indirectly act as an antifibrotic agent, as evidenced by our study results. If human studies provide fruitful results in the future, giloy could be exploited as a wonder drug for OSMF management.

## Figures and Tables

**Figure 1 materials-14-03374-f001:**
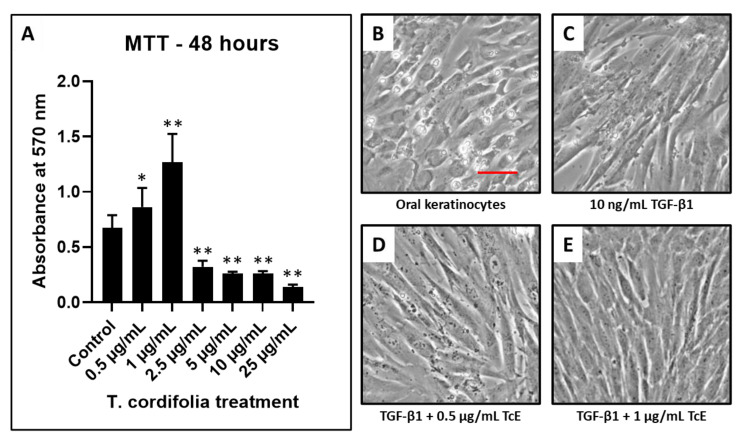
Cell viability by MTT assay and fibrosis induction in oral keratinocytes and TcE treatment. (**A**) MTT assay was performed to assess the viability of oral keratinocytes treated with various concentrations of TcE (0.5 µg/mL, 1 µg/mL, 2.5 µg/mL, 5 µg/mL, 10 µg/mL, and 25 µg/mL). (**B**,**C**) Photomicrographs of oral keratinocytes before and after fibrosis induction with TGF-β1. (**D**,**E**) Photomicrographs of fibrosis-induced oral keratinocytes treated with 0.5 µg/mL and 1 µg/mL of TcE. Scale bar = 100 µg/mL. TcE: *T. cordifolia* aqueous extract, TGF-β1: Transforming growth factor-beta 1. * *p* < 0.05, ** *p* < 0.001.

**Figure 2 materials-14-03374-f002:**
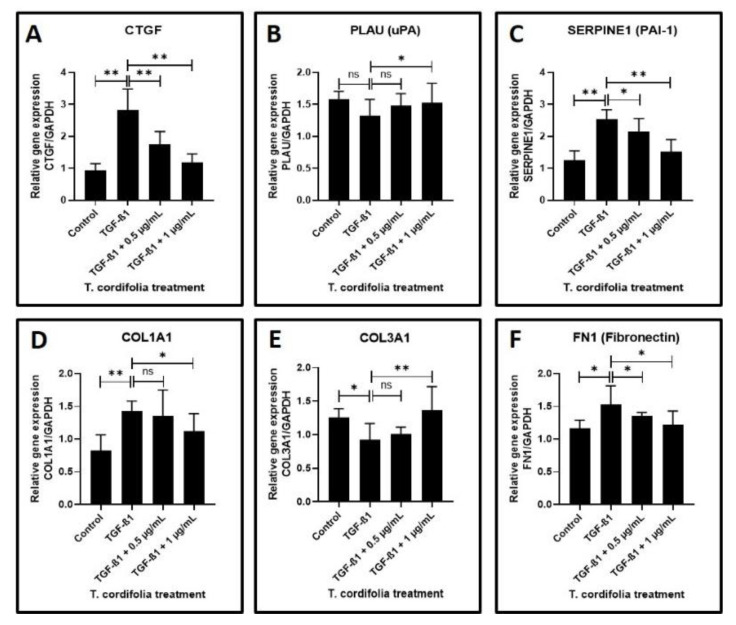
Analysis of gene expression by quantitative RT-PCR. (**A**) Fibrosis was induced by TGF-β1 in the oral keratinocytes and comparative gene expression analysis was carried out in the fibrosis-induced oral keratinocytes treated with various concentrations of TcE for matricellular protein-related gene expression for CTGF. (**B**,**C**) Fibrosis was induced by TGF-β1 in the oral keratinocytes and comparative gene expression analysis was carried out in the fibrosis-induced oral keratinocytes treated with various concentrations of TcE for plasmin system regulator protein-related gene expression for PLAU and SERPINE1. (**D**–**F**) Fibrosis was induced by TGF-β1 in the oral keratinocytes and comparative gene expression analysis was carried out in the fibrosis-induced oral keratinocytes treated with various concentrations of TcE for structural matrix protein-related gene expression for COL1A1, COL3A1, and FN1. ns—not significant, * *p* < 0.05, ** *p* < 0.001. CTGF: CCN2 or connective tissue growth factor, PLAU: Plasminogen activator, urokinase (µPA), SERPINE1: plasminogen activator inhibitor 1 (PAI-1), COL1A1: Collagen type I alpha-1, COL3A1: Collagen type III alpha-1, FN1: Fibronectin-1.

**Figure 3 materials-14-03374-f003:**
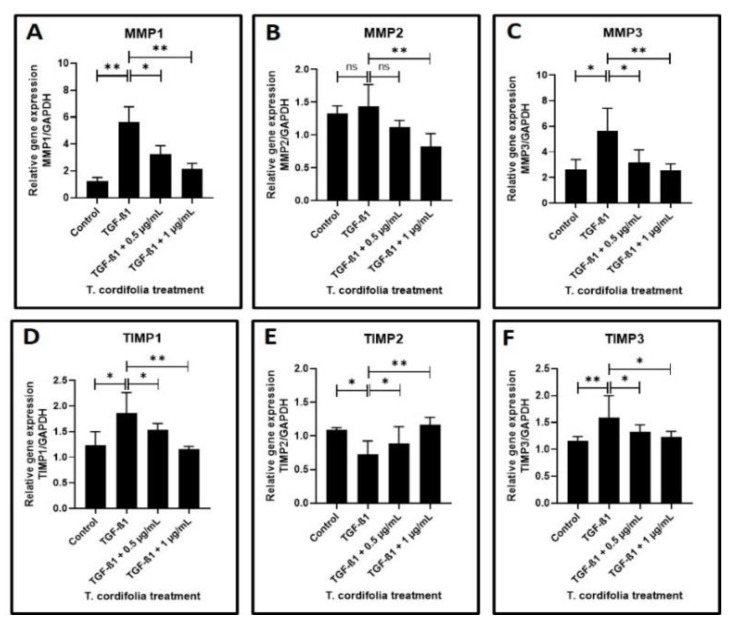
Analysis of gene expression by quantitative RT-PCR. (**A**–**C**) Fibrosis was induced by TGF-β1 in the oral keratinocytes and comparative gene expression analysis was carried out in the fibrosis-induced oral keratinocytes treated with various concentrations of TcE for matrix metalloproteinase-related gene expression for MMP1, MMP2, and MMP3. (**D**–**F**) Fibrosis was induced by TGF-β1 in the oral keratinocytes and comparative gene expression analysis was carried out in the fibrosis-induced oral keratinocytes treated with various concentrations of TcE for matrix-metalloproteinase inhibitor-related gene expression for TIMP1, TIMP2, and TIMP3. ns—not significant, * *p* < 0.05, ** *p* < 0.001. MMP1: Matrix metallopeptidase 1, MMP2: Matrix metallopeptidase 2, MMP3: Matrix metallopeptidase 3, TIMP1: Metallopeptidase inhibitor 1, TIMP2: Metallopeptidase inhibitor 2, TIMP3: Metallopeptidase inhibitor 3.

**Table 1 materials-14-03374-t001:** List of primers.

Gene	Forward Primer	Reverse Primer
TIMP1	5′-GGA GAG TGT CTG CGG ATA CTT C-3′	5′-GCA GGT AGT GAT GTG CAA GAG TC-3′
TIMP2	5′-ACC CTC TGT GAC TTC ATC GTG C-3′	5′-GGA GAT GTA GCA CGG GAT CAT G-3′
TIMP3	5′-TAC CGA GGC TTC ACC AAG ATG C-3′	5′-CAT CTT GCC ATC ATA GAC GCG AC-3′
MMP1	5′-ATG AAG CAG CCC AGA TGT GGA G-3′	5′-TGG TCC ACA TCT GCT CTT GGC A-3′
MMP2	5′-AGC GAG TGG ATG CCG CCT TTA A-3′	5′-CAT TCC AGG CAT CTG CGA TGA G-3′
MMP3	5′-CAC TCA CAG ACC TGA CTC GGT T-3′	5′-AAG CAG GAT CAC AGT TGG CTG G-3′
CTGF	5′-CTT GCG AAG CTG ACC TGG AAG A-3′	5′-CCG TCG GTA CAT ACT CCA CAG A-3′
PLAU	5′-GGC TTA ACT CCA ACA CGC AAG G-3′	5′-CCT CCT TGG AAC GGA TCT TCA G-3′
SERPINE1	5′-CTC ATC AGC CAC TGG AAA GGC A-3′	5′-GAC TCG TGA AGT CAG CCT GAA AC-3′
COL1A1	5′-GAT TCC CTG GAC CTA AAG GTG C-3′	5′-AGC CTC TCC ATC TTT GCC AGC A-3′
COL3A1	5′-TGG TCT GCA AGG AAT GCC TGG A-3′	5′-TCT TTC CCT GGG ACA CCA TCA G-3′
FN1	5′-ACA ACA CCG AGG TGA CTG AGA C-3′	5′-GGA CAC AAC GAT GCT TCC TGA G-3′
GAPDH	5′-GTC TCC TCT GAC TTC AAC AGC G-3′	5′-ACC ACC CTG TTG CTG TAG CCA A-3′

## Data Availability

The data presented in this study are available on request from the corresponding author.

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
