# Peer review of "Potential Application of an Aqueous Extract of Tinospora Cordifolia (Thunb.) Miers (Giloy) in Oral Submucous Fibrosis—An In Vitro Study"

_materials, 2021, doi:10.3390/ma14123374_

Round 1
Reviewer 1 Report
This paper aims to describe the antifibrotic effects of an extract of Tinospora Cordifolia in oral submucous fibrosis. The paper look as the initial steps of a project. The data provided in the paper are reduced to some MTT studies and the mRNA levels of several genes related to the fibrotic process. The novelty of the work resides in the employment of this extract to analyse its antifibrotic effect in primary cells.
Major comments:
- The effects found in this work are attributed to a whole extract from a plant. This procedure facilitates the obtention of the extract, but it also complicates the possibility of attributing the findings to a specific component of the extract (steroids, glycosides, phenolics, sesquiterpenoid, alkaloids, polysaccharides, diterpenoid lactones or aliphatic compounds). Perhaps the author could make a more comprehensive extraction process so he could separate the main components and have this chance.
- All along the paper, the author combines equally the units “µM” and “µg/mL” to refer to the same treatments. Those units do not represent the same amount. As this referee have been able to understand, the correct unit for this work is “µg/ml”. The cells have been treated with 0.5 and 1 µg/ml of the extract (not 0.5 and 1µM as mentioned).
- MTT studies are sensitive to redox process (as the coloured compound is produced by the mitochondria). The viability data should be confirmed employing another technique (for example neutral red).
- Figure 1A and Table 2 refers to the same data. In Figure 1 the data are represented in a graph whereas in Table 2 are presented the crude numbers.
- In Table 3 to Table 7 it is not clear if the data refers to the mRNA levels of the genes or protein levels. If it is the first case, then those data are already presented in figures 2 (A-F) and figure 3 (A-F). If they represent the protein levels of the genes, the author should explain how the data have been obtained (ELISA, Western blot) in the Materials and methods section.
- The author mention in Page 4 line 145 that “Since concentrations 0.5 and 1 μM reduced epithelial apoptosis”, and in page 9 line 221“Apoptosis of epithelial cells” : The author has not provided any evidence that the apoptotic process is involved (for example caspase activity, DNA fragmentation, PARP cleavage).
- Figure 1 (B-E). The images do not provide significant information. The author could quantified some parameter.
- If the purpose of the study is to analyse the antifibrotic effect of this plant, the protein levels of these genes should be determined. The changes in the mRNA levels presented are relevant, but insufficient.
Minor points:
- Page 7 line 198: “At 1 μM, TcE induced a significant reduction in expression of COL1A1 and decreased expression of COL3A1 compared with TGF-β1–induced cells”.
It should be “At 1 µg/ml, TcE induced a significant reduction in expression of COL1A1 and increased expression of COL3A1 compared with TGF-β1–induced cells”.
- Page 9 line 214 “itstransformation” should be separated.
- Page 10 line 275: “0.5and” should be separated.
- Page 10 line 277: “alone..” . There are 2 dots.
- Some expression should be rephrased: “Keratinocytes on treatment with”
Author Response
Reviewer 1:
This paper aims to describe the antifibrotic effects of an extract of Tinospora Cordifolia in oral submucous fibrosis. The paper look as the initial steps of a project. The data provided in the paper are reduced to some MTT studies and the mRNA levels of several genes related to the fibrotic process. The novelty of the work resides in the employment of this extract to analyse its antifibrotic effect in primary cells.
Major comments:
- The effects found in this work are attributed to a whole extract from a plant. This procedure facilitates the obtention of the extract, but it also complicates the possibility of attributing the findings to a specific component of the extract (steroids, glycosides, phenolics, sesquiterpenoid, alkaloids, polysaccharides, diterpenoid lactones or aliphatic compounds). Perhaps the author could make a more comprehensive extraction process so he could separate the main components and have this chance.
Response: The desired effects of plant extracts are largely attributed to the synergistic action of different phytomolecules present in the particular plant part. Separate phytocomponents from T. cordifolia or commonly found in other plants are being investigated extensively. Our idea was to investigate the effect of the plant part as a whole. In our future studies, we have planned to determine various phytocomponents, compare the effects with each individual molecule or combination of two or more molecules to conclusively and thoroughly make the further investigations on therapeutic potential of the plant extract.
- All along the paper, the author combines equally the units “µM” and “µg/mL” to refer to the same treatments. Those units do not represent the same amount. As this referee have been able to understand, the correct unit for this work is “µg/ml”. The cells have been treated with 0.5 and 1 µg/ml of the extract (not 0.5 and 1µM as mentioned).
Response: It was a mistake to write ““µM” instead of “µg/mL”. We have corrected the mistake wherever required.
- MTT studies are sensitive to redox process (as the coloured compound is produced by the mitochondria). The viability data should be confirmed employing another technique (for example neutral red).
Response: We have carefully followed the standard and well established protocol and also, we have taken care of the sensitivity was not compromised during any step due to external factors.
- Figure 1A and Table 2 refers to the same data. In Figure 1 the data are represented in a graph whereas in Table 2 are presented the crude numbers.
Response: We have removed the table 2 as it is not required.
- In Table 3 to Table 7 it is not clear if the data refers to the mRNA levels of the genes or protein levels. If it is the first case, then those data are already presented in figures 2 (A-F) and figure 3 (A-F). If they represent the protein levels of the genes, the author should explain how the data have been obtained (ELISA, Western blot) in the Materials and methods section.
Response: Table 3 to Table 7 represent mRNA levels for the genes expressing the respective proteins. We have removed the data tables as they are not required.
- The author mention in Page 4 line 145 that “Since concentrations 0.5 and 1 μM reduced epithelial apoptosis”, and in page 9 line 221“Apoptosis of epithelial cells” : The author has not provided any evidence that the apoptotic process is involved (for example caspase activity, DNA fragmentation, PARP cleavage).
Response: It was a mistake writing “toxicity” as “apoptosis” in “Since concentrations 0.5 and 1 μg/mL reduced epithelial apoptosis”. We have changed the sentence. We have removed “Apoptosis of epithelial cells” because we have not focused on apoptosis in this study.
- Figure 1 (B-E). The images do not provide significant information. The author could quantified some parameter.
Response: We have given the images to provide a brief idea on morphological changes occurring in the cells after the different treatment modalities. At this stage, we were unable to conclude any inference from these images but we have quantified the effects in terms of gene expression analyses.
- If the purpose of the study is to analyse the antifibrotic effect of this plant, the protein levels of these genes should be determined. The changes in the mRNA levels presented are relevant, but insufficient.
Response: This study was performed to see if the crude plant extract can exert the same effect rather better as individual phytomolecules as documented in the literature. In our future studies we will compare this effect with the present data and additional comparative experimentation. After only we will perform the protein quantification to determine the desired effect. This preliminary data has given us the insights about the experimental design and it will save us a lot of time and additional work being done without any significance.
Minor points:
- Page 7 line 198: “At 1 μM, TcE induced a significant reduction in expression of COL1A1 and decreased expression of COL3A1 compared with TGF-β1–induced cells”.
It should be “At 1 µg/ml, TcE induced a significant reduction in expression of COL1A1 and increased expression of COL3A1 compared with TGF-β1–induced cells”.
Response: Implemented the changes.
- Page 9 line 214 “itstransformation” should be separated.
Response: Separated its and transformation.
- Page 10 line 275: “0.5and” should be separated.
Response: Separated 0.5 and and.
- Page 10 line 277: “alone..” . There are 2 dots.\
Response: Removed a dot.
- Some expression should be rephrased: “Keratinocytes on treatment with”
Response: Rephrased the sentence.
Reviewer 2 Report
Nice work!
The scientific experiments seem ok.
On the other hand, there are a few typing mistakes in the text, and punctuation needs review.
Line 111: at room temperature?
Line 114: remove “was done.”
Some paragraphs in the discussion could be smaller and clearer, to become easier to read and evaluate data. I suggest that you write short paragraphs, and avoid superlative words, your results speak for you. I highlighted below just some parts that can be improved.
Line 214: separate “itstransformation”
Line 216: remove S from “results”
Line 221: remove on from “on the epithelial…”
Line 222: neoplastic
Line 224: remove ” has there been”
Line 236: review this part of this paragraph “It was observed that TGF-β1–induced oral keratinocytes underwent a plethora of molecular and biochemical changes stated below and thereafter. Concerning growth factors, there was a significant increase in the connective tissue growth factor expression following treatment with TGF-β1. This finding is of paramount importance as CTGF is one of the most implicated molecules in the pathobiology of OSMF where it acts as a mediator of fibrosis. The coarse particles of areca nut injure the oral mucosa and cause increased production of CTGF by oral cells at the sites of injury. CTGF has further been shown to induce collagen synthesis by both TGF-β1-dependent and independent pathways.”
This part of the conclusion could be added to the discussion
Line 287: The extracts of this herb have been found to have antioxidant activity [54] and have been found to exert immunostimulatory effects [55]. There is also evidence that giloy extracts could inhibit NF kappa B expression [56].
Author Response
Reviewer 2:
Nice work!
The scientific experiments seem ok.
On the other hand, there are a few typing mistakes in the text, and punctuation needs review.
Line 111: at room temperature?
Response: Yes. Added room before temperature.
Line 114: remove “was done.”
Response: Removed was done and rephrased the sentence.
Some paragraphs in the discussion could be smaller and clearer, to become easier to read and evaluate data. I suggest that you write short paragraphs, and avoid superlative words, your results speak for you. I highlighted below just some parts that can be improved.
Response: the paragraphs have been shortened and highlighted in yellow.
Line 214: separate “itstransformation”
Response: Separated its and transformation.
Line 216: remove S from “results”
Response: Removed S from “results”.
Line 221: remove on from “on the epithelial…”
Response: Removed “on”.
Line 224: remove ” has there been”
Response: Removed “has there been”.
Line 236: review this part of this paragraph “It was observed that TGF-β1–induced oral keratinocytes underwent a plethora of molecular and biochemical changes stated below and thereafter. Concerning growth factors, there was a significant increase in the connective tissue growth factor expression following treatment with TGF-β1. This finding is of paramount importance as CTGF is one of the most implicated molecules in the pathobiology of OSMF where it acts as a mediator of fibrosis. The coarse particles of areca nut injure the oral mucosa and cause increased production of CTGF by oral cells at the sites of injury. CTGF has further been shown to induce collagen synthesis by both TGF-β1-dependent and independent pathways.”
Response: the paragraph has been rephrased and highlighted.
In the present study the oral keratinocytes were induced with TGF-β1 prior to treatment with the extracts. Treatment of the cells with TGF-β1 caused several molecular and biochemical changes. Also there was a significant increase in the expression of connective tissue growth factor that plays a vital role in the pathogenesis of OSMF. It is a well-known fact that in the pathogenesis of areca nut induced OSMF the coarse particles of areca nut injure the oral mucosa and cause increased production of CTGF by oral cells at the sites of injury. CTGF has further been shown to induce collagen synthesis by both TGF-β1-dependent and -independent pathways. CTGF and TGF-β1 when acting in synchrony are known to cause an intense fibrotic response [50].
This part of the conclusion could be added to the discussion
Line 287: The extracts of this herb have been found to have antioxidant activity [54] and have been found to exert immunostimulatory effects [55]. There is also evidence that giloy extracts could inhibit NF kappa B expression [56].
Response: The same has been done and highlighted
Also studies have reported that the extracts of this herb have been found to have antioxidant activity [54] and have been found to exert immunostimulatory effects [55]. There is also evidence that giloy extracts could inhibit NF kappa B expression [56] all of which depict the antifibrotic activity of the herb.
Round 2
Reviewer 1 Report
The authors have properly answered to the questions formulated by this referee. The revised version of the manuscript has been improved.
As the author mentions, it is a preliminary study that will be followed in the future.